# Structural basis for oligoclonal T cell recognition of a shared p53 cancer neoantigen

Daichao Wu[1,2,3], D. Travis Gallagher[1,4], Ragul Gowthaman [1,3], Brian G. Pierce [1,3] & Roy A. Mariuzza[1,3 ✉]

Adoptive cell therapy (ACT) with tumor-specific T cells can mediate cancer regression. The main target of tumor-specific T cells are neoantigens arising from mutations in self-proteins. Although the majority of cancer neoantigens are unique to each patient, and therefore not broadly useful for ACT, some are shared. We studied oligoclonal T-cell receptors (TCRs) that recognize a shared neoepitope arising from a driver mutation in the p53 oncogene (p53R175H) presented by HLA-A2. Here we report structures of wild-type and mutant p53–HLA-A2 ligands, as well as structures of three tumor-specific TCRs bound to p53R175H–HLA-A2. These structures reveal how a driver mutation in p53 rendered a self-peptide visible to T cells. The TCRs employ structurally distinct strategies that are highly focused on the mutation to discriminate between mutant and wild-type p53. The TCR–p53R175H–HLA-A2 complexes provide a framework for designing TCRs to improve potency for ACT without sacrificing specificity.

[1] W.M. Keck Laboratory for Structural Biology, University of Maryland Institute for Bioscience and Biotechnology Research, Rockville, MD 20850, USA. [2] Department of Histology and Embryology, Hengyang Medical College, University of South China, Hengyang, Hunan 421001, China. [3] Department of Cell Biology and Molecular Genetics, University of Maryland, College Park, MD 20742, USA. [4] National Institute of Standards and Technology, Gaithersburg, MD 20899, USA. ✉email: rmariuzz@umd.edu

Adoptive cell therapy (ACT) with ex vivo-expanded tumor-infiltrating lymphocytes (TILs) can mediate durable cancer regression in patients with metastatic melanoma, cervix, bile duct, colon, and breast cancers[1–5]. This therapeutic effect is mediated mainly by CD8+ cytotoxic T cells[6] but CD4+ T cells are also likely to contribute[5]. The prime target of tumor-specific T cells are neoantigens that arise as a consequence of DNA alterations during malignant transformation[7]. Recent technological advances in mass spectrometry and high-throughput T cell-based assays have greatly accelerated the identification of neoantigens resulting from somatic mutations, as well as the T cells that recognize them, in individual patients.

A major challenge in developing broadly applicable neoantigen-directed ACT is the unique neoantigen repertoire of each cancer patient[7]. There are few shared mutated targets among patients, even among patients with similar cancers. For example, in a study of patients with gastrointestinal cancers, 99% of neoantigenic determinants recognized by neoantigen-reactive TILs were unique (private) and not shared (public) between any two patients[8]. Nevertheless, it has been possible to identify a limited number of shared cancer neoantigens. Of particular interest are neoantigens derived from oncogenes bearing driver mutations because these mutations are tumor-specific, biologically important for tumor progression, and likely to be expressed by all tumor cells[9]. In a seminal study of ACT, a patient with metastatic colorectal cancer was treated effectively with four distinct CD8+ T cell clones that specifically targeted a neoepitope arising from the KRAS G12D driver mutation in an HLA-C*08:02-restricted manner[2].

Other shared mutated neoantigens expressed by cancers of unrelated patients have now been identified. TP53, which encodes the tumor suppressor p53, is the most frequently mutated gene across all cancer types[10]. Indeed, TP53 mutations are found in 40–50% of cancer patients, and effect most of the hallmarks of cancer cells, including genomic instability, proliferation, and metastasis[11,12]. Mutant p53 predisposes to cancer development and is associated with ineffective therapeutic responses and unfavorable prognoses[10]. Despite these effects, no drug to abrogate the oncogenic functions of mutant p53 has yet been approved for any cancer treatment. A substantial portion of TP53 mutations occur at hotspot positions R175, G245, R248, R249, R273, and R282[10]. Because mutations at these sites confer a growth advantage to tumor cells and are associated with malignant progression, they are attractive candidates for targeted immunotherapy.

The immunogenicity of p53 mutations in patients with cancer was recently demonstrated by the detection of T responses against several shared p53 neoantigens, notably R175H and R248W[13,14]. Both of these driver mutations are located in the DNA-binding domain of p53 and alter its DNA-binding capacity[10]. Several TCRs have been isolated from TILs of epithelial cancer patients that recognize a neoepitope corresponding to residues 168–176 of p53R175H. This neoepitope includes the arginine-to-histidine mutation at position 175 (HMTEVVR**H**C)[13,14]. The TCRs are restricted by the common MHC class I allele HLA-A*02:01. These oligoclonal TCRs, transduced at high frequency into a patient's peripheral blood lymphocytes for ACT, may prove effective in eliminating tumors expressing HLA-A*02:01 and the p53R175H mutation[13,14].

With the goal of understanding TCR recognition of cancer neoantigens at the atomic level, we determined crystal structures of three p53R175H-specific TCRs (12-6, 38-10, and 1a2)[13,14] in complex with HLA-A*02:01 and the shared neoantigen p53R175H. In previous studies, we determined structures of human melanoma-specific TCRs bound to a neoepitope from mutant triose phosphate isomerase (mutTPI) and HLA-DR1[15,16]. However, that neoepitope, unlike KRAS G12D[2] or p53R175H[10], was only expressed in a single melanoma patient. The unique, rather than shared, nature of mutTPI, a feature that also characterizes the vast majority of cancer neoantigens discovered to date[7,8], precludes broad use of mutTPI-specific or similar TCRs in ACT. Structures have also been reported of TCRs in complex with epitopes from the tumor-associated antigens NY-ESO-1 and MART-1 bound to HLA-A2[17,18]. However, NY-ESO-1 and MART-1, unlike mutTPI[15] or p53R175H[13,14], are not neoantigens, but instead non-mutated self-antigens that are selectively expressed in certain cancer types. By contrast, the TCR–p53R175H–HLA-A2 structures described here involve a shared cancer neoantigen. The structures reveal how oligoclonal TCRs 12-6, 38-10, and 1a2 discriminate between wild-type and mutated p53, and demonstrate that there are multiple distinct solutions to recognizing the p53R175H neoepitope with sufficient affinity to mediate tumor cell killing.

## Results

**TCRs are highly specific for mutant p53 peptide.** TCRs 12-6, 38-10, and 1a2 were isolated by screening TILs from patients with metastatic colorectal cancer for reactivity towards mutated p53 neoantigens[13,14]. These HLA-A*0201-restricted TCRs recognize the p53R175H neoepitope using completely different $\alpha/\beta$ chain pairs (Table 1). 12-6 utilizes gene segments TRAV12-1 and TRAJ13 for the $\alpha$ chain and TRBV6-1 and TRBJ2-7 for the $\beta$ chain; 38-10 utilizes TRAV38-1 and TRAJ28 for the $\alpha$ chain and TRBV10-3 and TRBJ1-6 for the $\beta$ chain; and 1a2 utilizes TRAV12-3 and TRAJ12 for the $\alpha$ chain and TRBV27 and TRBJ2-3 for the $\beta$ chain[13,14]. We used surface plasmon resonance (SPR) to measure the affinity of TCRs 12-6, 38-10, and 1a2 to HLA-A2 loaded with wild-type or mutant p53 peptide (Fig. 1). Recombinant TCR and pMHC proteins were expressed by in vitro folding from bacterial inclusion bodies. Biotinylated wild-type p53–HLA-A2 or p53R175H–HLA-A2 was directionally coupled to a streptavidin-coated biosensor surface and different concentrations of 12-6, 38-10, or 1a2 were flowed sequentially over the immobilized pMHC ligand. We detected no apparent interaction between any of these TCRs and wild-type p53–HLA-A2, even after injecting high concentrations of TCR (up to 328 μM for 12-6, 232 μM for 38-10, and 153 μM for 1a2) (Fig. 1a–c). By contrast, these TCRs bound mutant p53R175H–HLA-A2 with dissociation constants ($K_D$s) of 1.1 μM for 12-6, 39.9 μM for 38-10, and 16.2 μM for 1a2 (Fig. 1d–f). These affinities are well within the range of TCRs specific for microbial or other foreign antigens ($K_D = 1$–50 μM), but substantially higher than the affinities of autoimmune TCRs that recognize non-mutated self-antigens

**Table 1 Neoepitope p53R175H-reactive TCR germline genes and CDR3 sequences.**

| Name | TRAV | TRAJ | CDR3$\alpha$ | TRBV | TRBJ | CDR3$\beta$ | Reference |
|------|------|------|--------------|------|------|-------------|-----------|
| 12-6 | 12-1 | 13 | CVVQPGGYQKVTF | 6-1 | 2-7 | CASSEGLWQVGDEQYF | 14 |
| 38-10 | 38-1 | 28 | CAFMGYSGAGSYQLTF | 10-3 | 1-6 | CAISELVTGDSPLHF | 14 |
| 1a2 | 12-3 | 12 | CAMSGLKEDSSYKLIF | 27 | 2-3 | CASSIQQGADTQYF | 13 |

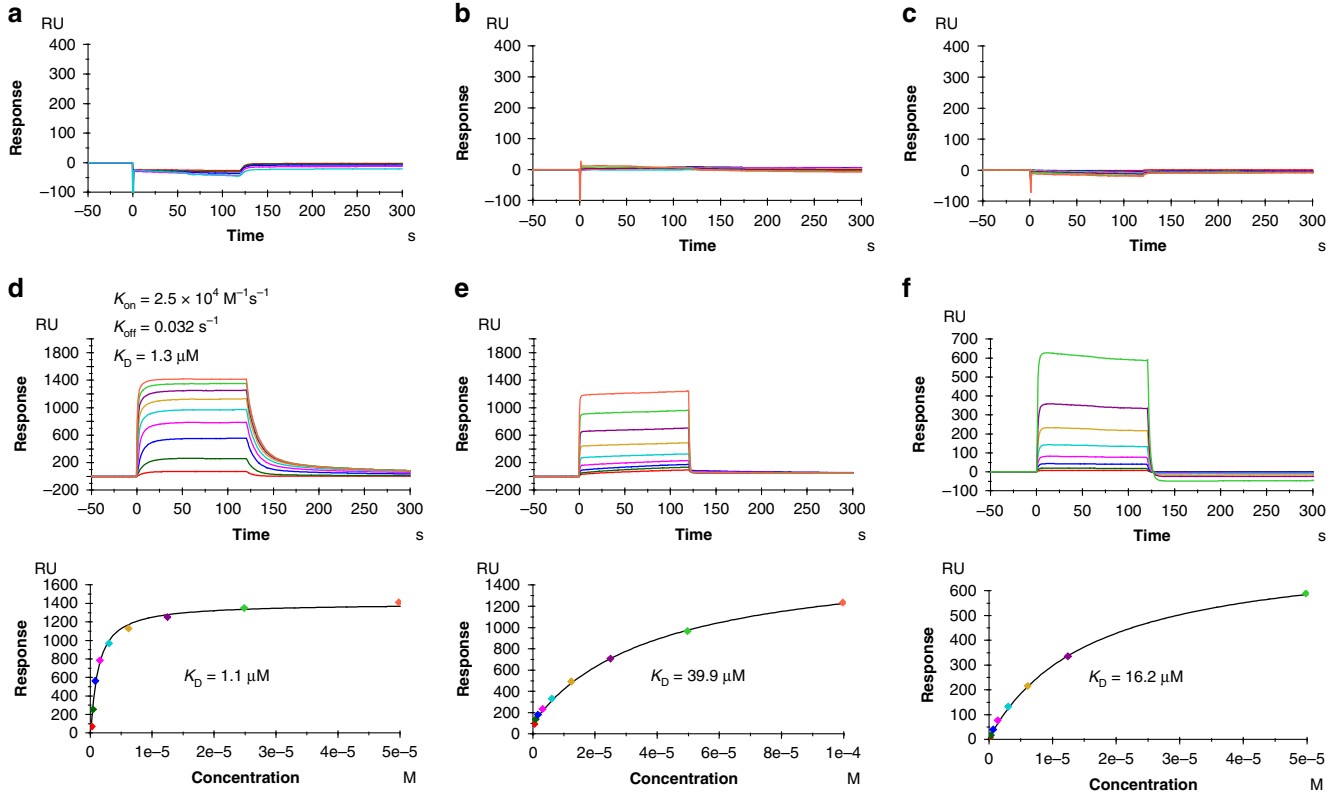

**Fig. 1 SPR analysis of TCR binding to p53–HLA-A2 and p53R175H–HLA-A2. a** TCR 12-6 at concentrations of 20.5, 41, 82, 164, and 328 μM was injected over immobilized p53–HLA-A2 (3000 RU). **b** TCR 38-10 at concentrations of 0.9, 1.8, 3.6, 7.3, 14.5, 29, 58, and 232 μM was injected over immobilized p53–HLA-A2. **c** TCR 1a2 at concentrations of 0.6, 1.2, 2.4, 4.8, 9.6, 19.1, 38.3, 76.5, and 153 μM was injected over immobilized p53–HLA-A2. **d** (upper) TCR 12-6 at concentrations of 0.19, 0.39, 0.78, 1.56, 3.12, 6.25, 12.5, 25, and 50 μM was injected over immobilized p53R175H–HLA-A2 (3000 RU). (lower) Fitting curve for equilibrium binding that resulted in a $K_D$ of 1.1 μM. **e** (upper) TCR 38-10 at concentrations of 0.39, 0.78, 1.56, 3.12, 6.25, 12.5, 25, 50.0, and 100 μM was injected over immobilized p53R175H–HLA-A2. (lower) Fitting curve for equilibrium binding that resulted in a $K_D$ of 39.9 μM. **f** (upper) TCR 1a2 at concentrations of 0.19, 0.39, 0.78, 1.56, 3.12, 6.25, 12.5, and 50 μM was injected over immobilized p53R175H–HLA-A2. (lower) Fitting curve for equilibrium binding that resulted in a $K_D$ of 16.2 μM. Source data are provided in a Source data file.

($K_D > 100$ μM)[19]. Kinetic parameters (on- and off-rates) for the binding of TCR 12-6 to p53R175H–HLA-A2 were $k_{on} = 2.5 \times 10^4$ $M^{-1}$ $s^{-1}$ and $k_{off} = 0.032$ $s^{-1}$, corresponding to a $K_D$ of 1.3 μM (Fig. 1d), which is in close agreement with the $K_D$ from equilibrium analysis (1.1 μM). For TCRs 38-10 and 1a2, $k_{on}$ and $k_{off}$ were too rapid to be measured reliably (Fig. 1e, f). The exquisite specificity of 12-6, 38-10, and 1a2 for p53R175H compared with p53, as measured by SPR, is consistent with functional assays showing that T cells transduced with these TCRs can be activated by APCs pulsed with subnanomolar concentrations of mutant p53R175H peptide, but do not respond to wild-type p53 peptide, even at >1000-fold higher concentrations[13,14].

**Differences between p53 and p53R175H are confined to mutation site.** To understand how the conservative arginine-to-histidine mutation in p53R175H, which replaces one positively charged amino acid by another, renders this peptide immunogenic, we determined the structures of the wild-type p53–HLA-A2 and mutant p53R175H–HLA-A2 complexes to 2.37 and 2.38 Å resolution, respectively (Supplementary Table 1) (Fig. 2a). Clear and continuous electron density extending along the entire length of both MHC-bound peptides allowed confident identification of all peptide atoms (Supplementary Fig. 1). In both p53–HLA-A2 and p53R175H–HLA-A2, the peptide is bound in conventional orientation with the side chains of P2 Met and P9 Cys accommodated in pockets B and F, respectively, in the peptide-binding groove (Fig. 2b). Methionine and cysteine are

among the most common residues at primary anchor positions P2 (Leu > Thr > Met ~ Val > Ile) and P9 (Val > Ile > Thr > Ala > Cys > Leu), and are known to confer high affinity for HLA-A*02:01[20]. The solvent-exposed side chains of P1 His, P4 Glu, P7 Arg, and P8 Arg/His project away from the peptide-binding groove and compose a highly featured surface for potential interactions with TCR. By contrast, the side chains of P5 Val and P6 Val present a relatively featureless, non-protruding surface that may be a difficult target for TCR recognition.

The p53–HLA-A2 and p53R175H–HLA-A2 complexes exhibit very little structural deviation from each other (Fig. 2a). In particular, the wild-type and mutant p53 peptides are highly superimposable, except at the P8 mutation site (Fig. 2b). The root-mean-square difference (r.m.s.d.) for α-carbon atoms in the peptide chains is 0.18 Å, while for all atoms, excluding the P8 Arg and His side chains, it is 0.45 Å. Therefore, structural differences between the p53–HLA-A2 and p53R175H–HLA-A2 complexes that disclosed the naturally altered p53 self-peptide to the T cells of cancer patients[13,14] are restricted to the mutation site at P8.

**TCRs are shifted toward C-terminus of p53R175 peptide.** To understand how TCRs 12-6, 38-10, and 1a2 discriminate between wild-type and mutant p53 epitopes with exquisite specificity (Fig. 1), we determined the structures of the 12-6–p53R175H–HLA-A2, 38-10–p53R175H–HLA-A2, and 1a2–p53R175H–HLA-A2 complexes to 2.61, 2.46, and 3.00 Å resolution (Supplementary Table 2) (Fig. 3). The interface

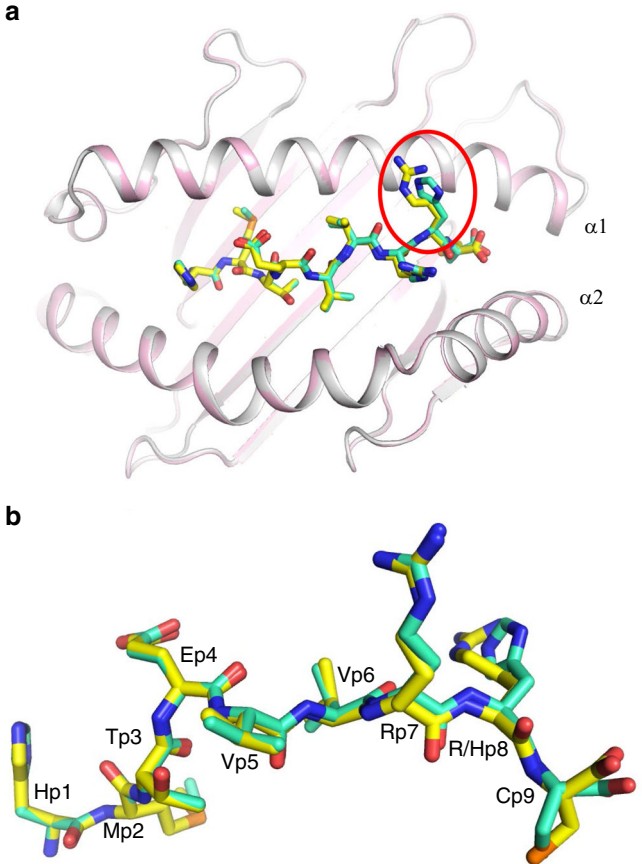

**Fig. 2 Conformation of wild-type and mutant p53 peptides bound to HLA-A2. a** Top view of superposed p53–HLA-A2 and p53R175H–HLA-A2 complexes. Wild-type and mutant p53 peptides are yellow and green, respectively. The P8 mutation site is circled. HLA-A2 is gray (p53–HLA-A2) or light pink (p53R175H–HLA-A2). **b** Side view of superposed wild-type and mutant p53 peptides. Residue labels are aligned with the α-carbon atom of the respective residue. Carbon atoms are yellow (p53) or green (p53R175H); nitrogen atoms are blue; oxygen atoms are red; sulfur atoms are orange.

between TCR and pMHC was in unambiguous electron density in all three complex structures (Supplementary Fig. 2). The 1a2–p53R175H–HLA-A2 crystal contains two complex molecules in the asymmetric unit. The r.m.s.d. in α-carbon positions for the TCR VαVβ and MHC α1α2 modules, including the p53R175H peptide, is 0.45 Å for the two 1a2–p53R175H–HLA-A2 complexes. Based on this close similarity, the following description of 1a2–p53R175H–HLA-A2 interactions applies to both molecules in the asymmetric unit.

TCRs 12-6, 38-10, and 1a2 dock over p53R175H–HLA-A2 in a canonical diagonal orientation, with Vα over the α2 helix of HLA-A2 and Vβ over the α1 helix, but with markedly different crossing angles of TCR to pMHC[21]: 51° for 12-6, 34° for 38-10, and 30° for 1a2 (Fig. 4a–c). The complexes also differ with respect to incident angle[22], which corresponds to the degree of tilt of TCR over pMHC: 20° for 12-6, 27° for 38-10, and 1° for 1a2. In comparison with TCR–pMHC class I complexes from the PDB (133 other complexes, 136 total) (Supplementary Table 4), the 38-10 TCR complex has the fourth-highest incident angle (97th percentile), and the 12-6 TCR complex has the 14th-highest (90th percentile). Of note, the three complexes with higher incident angles than the 38-10 TCR complex are two reversed-polarity TCRs[23] and a TCR in complex with a non-stimulatory peptide[24].

However, whereas the unusual docking geometries of these latter TCRs are incompatible with signaling, that of 38-10 allows a robust response to antigen[13].

All three TCRs are shifted towards the C-terminus of the p53R175H peptide, which is the site of the driver mutation at P8. To quantitate the shifts, we projected the positions of the TCR centers onto the pMHC plane, where the $x$-axis is aligned with the peptide and a more positive $x$ value indicates a shift toward the peptide C-terminus (Supplementary Table 4). Remarkably, TCR 38-10 exhibits the third-highest C-terminal shift among 136 TCR–pMHC class I structures reported to date, with 12-6 (22nd-highest) and 1a2 (26th-highest also quite shifted. Only two reversed-polarity TCRs[23] are more skewed along the peptide than 38-10. The C-terminal shift of TCRs 38-10, 12-6, and 1a2 is key to their ability to discriminate between wild-type and mutant p53 peptides (see below).

As depicted by the footprints of TCRs 12-6 and 1a2 on the pMHC surface (Fig. 4d, f), both TCRs, which were derived from different cancer patients[13,14], establish contacts with the C-terminal half of the p53R175H peptide mainly via the CDR3β loop. By contrast, TCR 38-10, which was isolated from the same patient as 12-6[13], engages the C-terminal half of the p53R175H peptide mostly through CDR3α (Fig. 4e). Overall, the footprints of 12-6 and 1a2 on pMHC resemble each other more closely than either footprint resembles that of 38-10.

**Vα dominates contacts with MHC.** TCRs 12-6, 38-10, and 1a2 engage HLA-A2 through distinct sets of interactions, despite some broad similarities (Supplementary Table 5) (Fig. 5). Of the total number of contacts (92) that TCR 12-6 makes with HLA-A2, excluding p53R175H, CDR1α, CDR2α, and CDR3α contribute 5%, 42%, and 30%, respectively, compared with 5%, 3%, and 13% for CDR1β, CDR2β, and CDR3β, respectively (Tables 2, 3). Hence, Vα dominates the interactions of 12-6 with MHC (72 of 92 contacts; 78%), with CDR2α accounting for more of the binding interface than any other CDR. This dominance of CDR2α in MHC contacts is unusual for natural TCR interactions: seven other MHC class I-restricted TCR–pMHC structures were identified with greater percentages of CDR2α contacts out of total TCR–MHC contacts, but these included interactions with non-natural peptides from yeast display (PDB codes 4N5E, 3TFK, 4MXQ, 4MVB, 4N0C), an engineered TCR with altered peptide specificity (PDB code 5E9D), and an engineered autoreactive TCR interacting with HLA-A1 and a titin-derived peptide (PDB code 5BS0).

12-6 relies on the somatically-generated CDR3 loops for MHC recognition to approximately the same extent as the germline-encoded CDR1 and CDR2 loops (47 versus 53 contacts) (Table 2). Residues Ser52α and Ser53α of CDR2α form a dense network of four hydrogen bonds with Glu154H of the HLA-A2 α2 helix (Supplementary Table 5) (Fig. 5b). By contrast, interactions with the HLA-A2 α1 helix are almost entirely somatically-encoded, with Trp98β of CDR3β mediating 15 hydrophobic contacts with Ala69H, Gln72H and Thr73H, and Gly94α and Gln96α of CDR3α forming hydrogen bonds with the R65H side chain (Fig. 5a).

TCR 38-10 makes less than half as many contacts with HLA-A2 as does TCR 12-6 (38 versus 92) (Table 2) (Fig. 5c, d). This imbalance is due to a nearly complete absence of interactions with the HLA-A2 α1 helix (Supplementary Table 5), as a consequence of the highly tilted binding mode of 38-10, which is characterized by a 27° incident angle of TCR over pMHC (see above). Of the total contacts between 38-10 and HLA-A2, CDR1α, CDR2α, and CDR3α account for 21%, 16%, and 37%, respectively, compared with 13%, 5%, and 8% for CDR1β, CDR2β, and CDR3β,

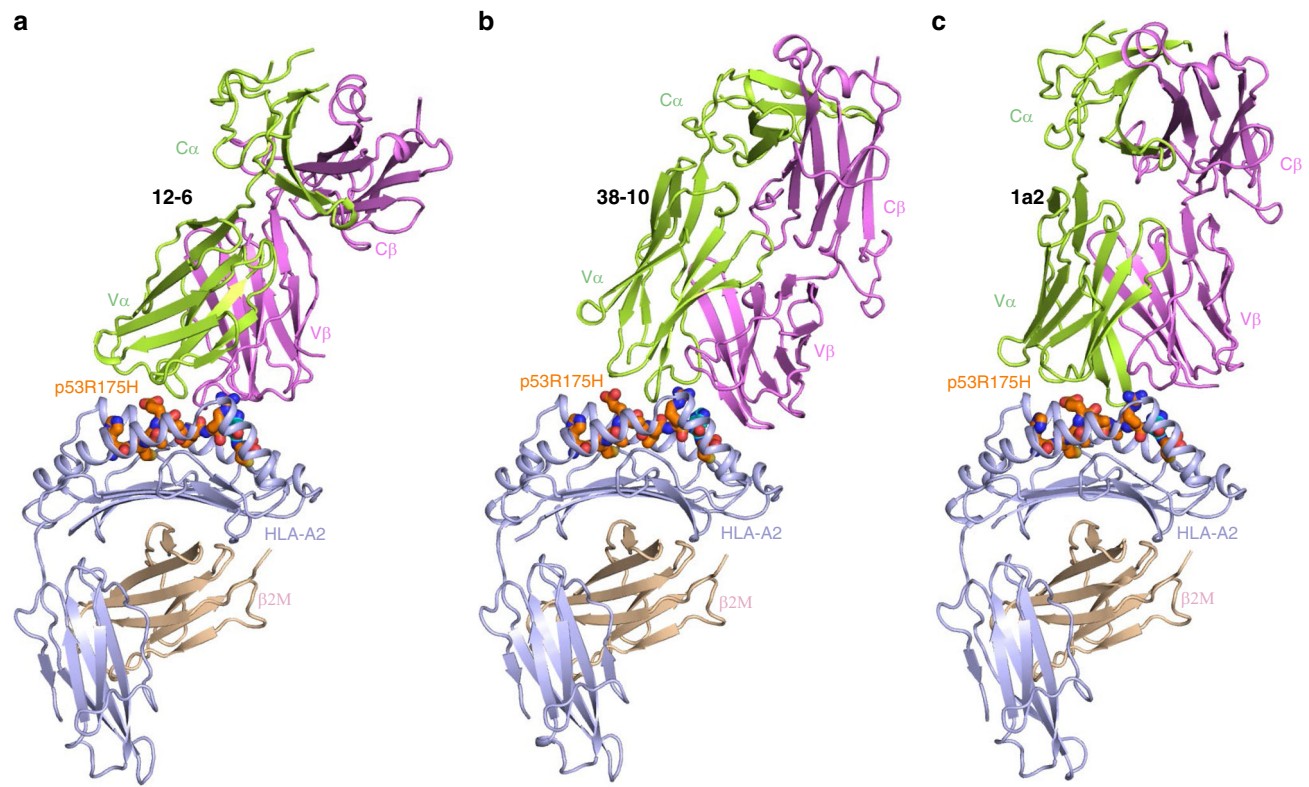

**Fig. 3 Structures of TCR–p53R175H–HLA-A2 complexes. a** Side view of 12-6–p53R175H–HLA-A2 complex (ribbon diagram). **b** 38-10–p53R175H–HLA-A2 complex. **c** 1a2–p53R175H–HLA-A2 complex. TCR $\alpha$ chain, green; TCR $\beta$ chain, violet; HLA-A2 heavy chain, light blue; $\beta_2$-microglobulin ($\beta_2$m), wheat. The p53R175H peptide is orange with the mutated P8 His residue highlighted in dark blue.

respectively. As in the case of 12-6, V$\alpha$ dominates the interactions with MHC (41 of 49 contacts; 84%). 38-10 engages the HLA-A2 $\alpha$2 helix using all three V$\alpha$ CDR loops. Thus, Asn31$\alpha$, Glu52$\alpha$, Lys55$\alpha$, and Tyr97$\alpha$ form a cluster of six hydrogen bonds with Ala150H, His151H, and Gln155H (Supplementary Table 5) (Fig. 5d).

Of the total contacts (53) with HLA-A2 made by TCR 1a2, CDR1$\alpha$, CDR2$\alpha$, and CDR3$\alpha$ contribute 22%, 30%, and 47%, respectively, compared with 0%, 0%, and 2% by CDR1$\beta$, CDR2$\beta$, and CDR3$\beta$ (Table 3). Thus, in the 1a2–p53R175H–HLA-A2 complex, V$\alpha$ accounts for 98% of contacts with MHC, even more than in the 12-6–p53R175H–HLA-A2 (78%) and 38-10–p53R175H–HLA-A2 (84%) complexes. The level of V$\alpha$ dominance in the TCR–MHC interactions of the 1a2 complex appears highly unusual, as it is only exhibited by one other MHC class I-restricted TCR, C1-28, which engages HLA-A24 and an HIV epitope (PDB code 3VXM). The paucity of V$\beta$–MHC interactions in all three complexes is in large measure attributable to the marked shift of the TCRs toward the C-terminus of the p53R175H peptide (Supplementary Table 4), which effectively disengages V$\beta$ from the MHC $\alpha$1 and $\alpha$2 helices, although not from the bound peptide (see below). The pronounced tilt (incident angle) of the 38-10 TCR compensates in part for this effect, resulting in more V$\beta$–MHC contacts than the other two TCRs.

TCR 1a2 relies heavily on the somatically-generated CDR3$\alpha$ loop for MHC recognition (Fig. 5e). Indeed, the percentage contribution of this loop to interactions with MHC (49% of contacts) exceeds that of any other CDR in any of the three complexes (Table 3). Five consecutive residues at the tip of the 1a2 CDR3$\alpha$ loop (Leu94$\alpha$–Ser98$\alpha$) pack tightly against the HLA-A2 $\alpha$1 helix, with two hydrogen bonds providing additional stabilization (Supplementary Table 5).

**TCRs target p53 driver mutation.** Upon binding p53R175H–HLA-A2, TCRs 12-6, 38-10, and 1a2 bury 71% (303 Å$^2$), 76% (336 Å$^2$), and 76% (304 Å$^2$), respectively, of the peptide solvent-accessible surface, which is typical for TCR–pMHC complexes[25]. However, the large majority of interactions between these TCRs and the p53R175H peptide involves C-terminal residues P7 Arg and P8 His: 44 of 52 van der Waals contacts and 7 of 9 hydrogen bonds for 12-6, 57 of 74 van der Waals contacts and 5 of 8 hydrogen bonds for 38-10, and 37 of 53 van der Waals contacts and 6 of 8 hydrogen bonds for 1a2 (Supplementary Table 6) (Fig. 6). These interactions are about evenly distributed between P7 Arg and P8 His, which suggests the functional importance of both residues for TCR binding. This conclusion is supported by binding energy calculations using Rosetta[26] to predict changes in TCR affinity upon alanine substitution of all peptide residues in the three complexes. In each case, the largest $\Delta\Delta G$ values, ranging from 1.7 to 3.6 kcal/mol, were observed for residues P7 and P8 (Supplementary Table 7). Therefore, all three TCRs focus on the C-terminal portion of the antigenic peptide for binding, in sharp contrast to most other TCRs, which preferentially target the central portion of peptides, corresponding to residues P4–P6[25]. The TCRs discriminate between mutant and wild-type p53 by minimizing interactions with the central and N-terminal portions of p53R175H, which are structurally identical in the wild-type peptide (Fig. 2b). Interactions between 12-6 and the p53R175H peptide are mediated almost exclusively by CDR3$\beta$ (Supplementary Table 6), whereas 38-10 and 1a2 employ both CDR3$\alpha$ and CDR$\beta$ for peptide recognition (Fig. 6a, b).

Consistent with differences in $\alpha/\beta$ chain pairing and docking geometry (Fig. 4), TCRs 12-6, 38-10, and 1a2 use distinct strategies to achieve highly specific recognition of the mutant p53 peptide relative to wild-type, as demonstrated by SPR (Fig. 1). However, the TCRs share a pronounced skewing toward the

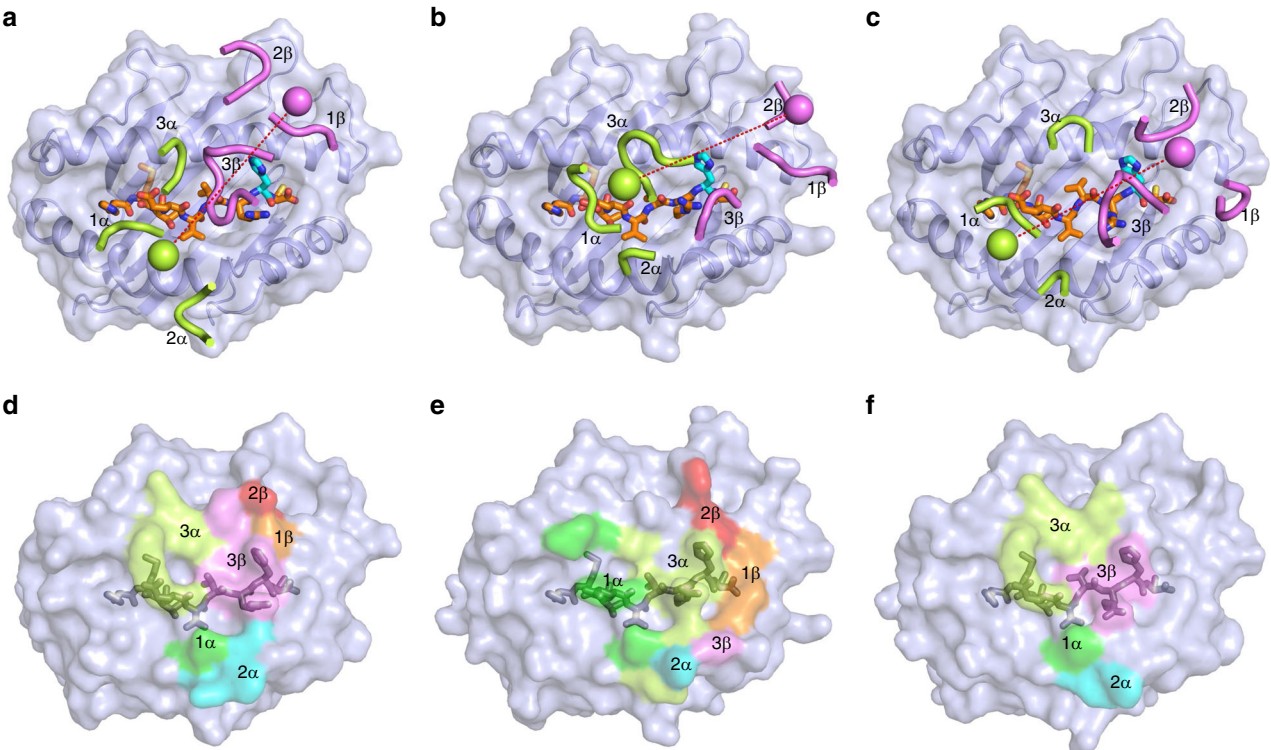

**Fig. 4 Comparison of TCR footprints on p53R175H–HLA-A2. a** Positions of CDR loops of TCR 12-6 on p53R175H–HLA-A2 (top view). CDRs of 12-6 are shown as numbered green (CDR1$\alpha$, CDR2$\alpha$, and CDR3$\alpha$) or violet (CDR1$\beta$, CDR2$\beta$, and CDR3$\beta$) loops. HLA-A2 is depicted as a light blue surface. The p53R175H peptide is drawn in orange in stick representation with the mutated P8 His residue in cyan. The green and violet spheres mark the positions of the conserved intrachain disulfide of the V$\alpha$ and V$\beta$ domains, respectively. The red dashed line indicates the crossing angle of TCR to pMHC. **b** Positions of CDR loops of TCR 38-10 on p53R175H–HLA-A2 (top view). **c** Positions of CDR loops of TCR 1a2 on p53R175H–HLA-A2 (top view). **d** Footprint of TCR 12-6 on p53R175H–HLA-A2. The top of the MHC molecule is depicted as a light blue surface. The areas contacted by individual CDR loops are color-coded: CDR1$\alpha$, dark green; CDR2$\alpha$, cyan; CDR3$\alpha$, light green; CDR1$\beta$, orange; CDR2$\beta$, red; CDR3$\beta$, pink. **e** Footprint of TCR 38-10 on p53R175H–HLA-A2. **f** Footprint of TCR 1a2 on p53R175H–HLA-A2.

peptide C-terminus (Supplementary Table 4) as a result of positioning one or both of their CDR3 loops directly over P8 His (Fig. 6a–c). In the unbound p53R175H–HLA-A2 structure (Fig. 2a), the P8 His imidazole ring has one face against the side chain of Val76H of the HLA-A2 $\alpha$1 helix, leaving its other face and most of its edge available for TCR binding. Each TCR provides close contacts that tightly sandwich the imidazole between Val76H and a specific CDR3 side chain: CDR3$\beta$ Gln99 in 12-6, CDR3$\alpha$ Tyr103 in 38-10, and CDR3$\beta$ Gln97 in 1a2 (Fig. 6c). In addition, the phenyl ring of CDR3$\alpha$ Tyr103 forms $\pi$–$\pi$ stacking interactions with the imidazole ring of P8 His. Further selectivity for the mutant p53 peptide arises from hydrogen bonds with the P8 His side chain: 12-6 Glu95$\beta$ O$\varepsilon$2–N$\delta$1 P8 His, 12-6 Trp98$\beta$ N$\varepsilon$1–N$\varepsilon$2 P8 His, 38-10 Tyr31$\beta$ OH–N$\delta$1 P8 His, and 1a2 Ser98$\alpha$ O$\gamma$–N$\varepsilon$2 P8 His (Supplementary Table 6).

To assess the effect of replacing P8 His by Arg, which corresponds to reversion to the wild-type p53 peptide, we performed in silico mutagenesis using Rosetta[27]. A similar modeling protocol was previously used to predict binding effects of TCR–pMHC interface mutations[28]. Peptide substitutions were modeled in each X-ray complex structure, followed by side-chain packing and energetics-based scoring to calculate $\Delta\Delta G$. Predicted $\Delta\Delta G$ values were 1.6, 1.2, and 2.0 Rosetta energy units (REU; analogous to kcal/mol) for 12-6, 38-10, and 1a2, respectively, suggesting substantial losses in TCR binding affinity for wild-type p53 peptide (Supplementary Table 7), as observed experimentally by SPR (Fig. 1). To assess possible structural defects leading to TCR affinity loss for the p53 revertant peptide, we calculated TCR–pMHC shape complementarity statistics ($S_c$) for the X-ray

and modeled p53 revertant interfaces. $S_c$ values for p53175H interfaces are 0.72, 0.64, and 0.69 for 12-6, 38-10, and 1a2, respectively, commensurate with other MHC class I TCR–pMHC structures in the TCR3d database[29], while they are 0.69, 0.57, and 0.71 for p53 revertants and the same respective TCRs, indicating loss of shape complementarity for the 12-6 and 38-10 TCR interfaces, and less predicted effect on the shape complementarity of the 1a2 interface. To further investigate the mechanistic basis of peptide specificity for these TCRs, the individual Rosetta scoring function terms comprising the predicted $\Delta\Delta G$ values noted above and in Supplementary Table 7 were obtained (Supplementary Table 8). This revealed that loss of favorable van der Waals interactions dominated the change in predicted binding affinity for the 12-6 TCR, whereas disruptions of side chain–side chain hydrogen bond interactions involving P8 His were primarily responsible for predicted 38-10 and 1a2 TCR affinity losses.

**Conformational changes optimize TCR–pMHC interactions.** Superposition of the MHC $\alpha$1$\alpha$2 domains of unbound p53R175H–HLA-A2 onto those of p53R175H–HLA-A2 in complexes with TCRs 12-6, 38-10, and 1a2 showed small yet relevant differences in peptide conformation, corresponding to r. m.s.d. of 0.76, 0.81, and 0.70 Å, respectively, for main-chain atoms of p53R175H. In each case, peptide residues P4–P8 are more deeply buried in the peptide-binding groove after TCR engagement (Fig. 7a–c), thereby enabling the TCR to maximize interactions with MHC. P6 Val underwent the largest individual displacement: 1.4, 1.6, and 0.9 Å in its $\alpha$-carbon position in the

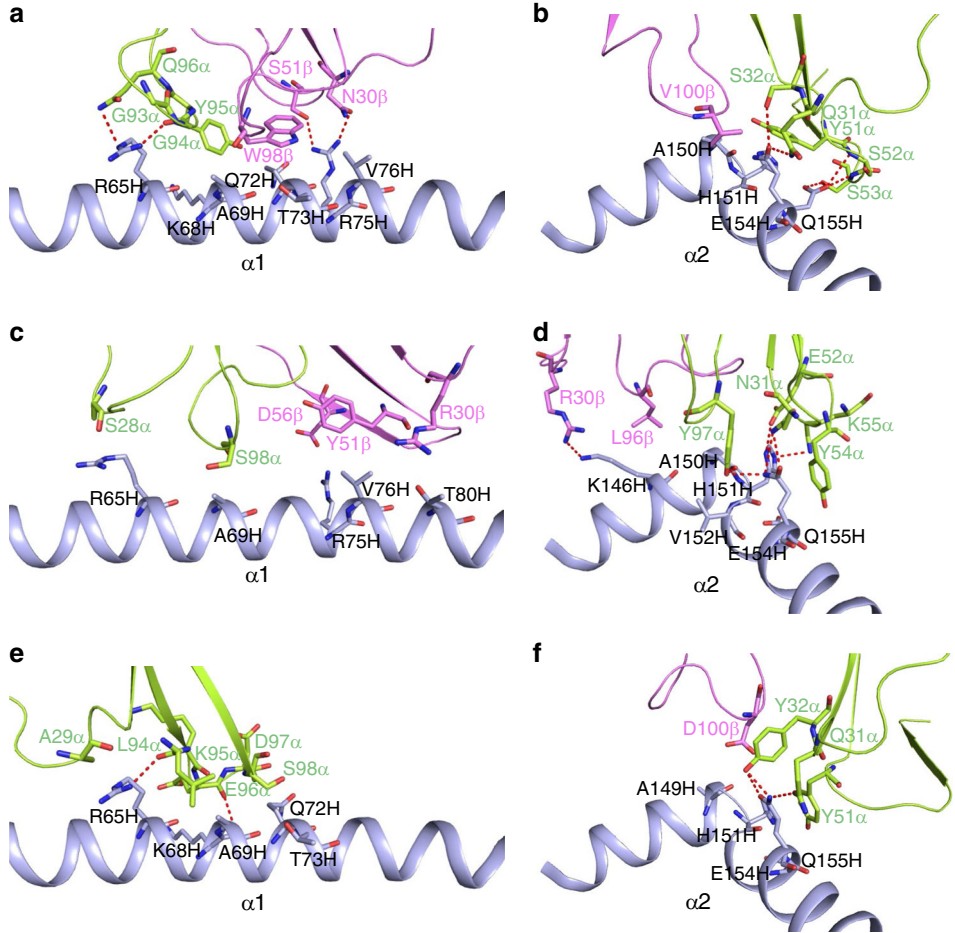

**Fig. 5 Interactions of TCRs with HLA-A2. a** Interactions between 12-6 and the HLA-A2 α1 helix. The side chains of contacting residues are drawn in stick representation with carbon atoms in green (TCR α chain), violet (TCR β chain) or light blue (HLA-A2), nitrogen atoms in dark blue, and oxygen atoms in red. Hydrogen bonds are indicated by red dashed lines. **b** Interactions between 12-6 and the HLA-A2 α2 helix. **c** Interactions between 38-10 and the HLA-A2 α1 helix. **d** Interactions between 38-10 and the HLA-A2 α2 helix. **e** Interactions between 1a2 and the HLA-A2 α1 helix. **f** Interactions between 1a2 and the HLA-A2 α2 helix.

**Table 2 TCR CDR atomic contacts with peptide and MHC (number of contacts).**

|  |  | α chain | | | | β chain | | | | |
|---|---|---|---|---|---|---|---|---|---|---|
|  |  | CDR1 | CDR2 | HV4 | CDR3 | CDR1 | CDR2 | HV4 | CDR3 | Total[a] |
| 12-6 | Peptide | 0 | 0 | 0 | 3 | 0 | 0 | 0 | 61 | 64 |
|  | MHC | 5 | 39 | 0 | 28 | 5 | 3 | 0 | 12 | 92 |
| 38-10 | Peptide | 12 | 0 | 0 | 54 | 11 | 1 | 0 | 4 | 82 |
|  | MHC | 8 | 6 | 0 | 14 | 5 | 2 | 0 | 3 | 38 |
| 1a2 | Peptide | 11 | 0 | 0 | 13 | 0 | 1 | 0 | 36 | 61 |
|  | MHC | 13 | 18 | 0 | 28 | 0 | 0 | 0 | 1 | 60 |

Contacts were calculated between non-hydrogen atoms with a 4.0 Å distance cutoff.
[a]Total contacts reflect the total number of TCR–MHC or TCR–peptide contacts.

12-6–p53R175H–HLA-A2, 38-10–p53R175H–HLA-A2, and 1a2–p53R175H–HLA-A2 complexes, respectively. Binding of TCR 38-10 induced structural adjustments in HLA-A2 at a bend in the α2 helix corresponding to residues 147–151, which underwent an average displacement of 1.3 Å in the position of their α-carbons toward the p53R175H peptide (Fig. 7b). There are two adjacent water molecules under peptide residues P6–P8 in the wild-type p53–HLA-A2 and unbound mutant p53R175H–HLA-A2 structures. These waters are retained in the 38-10–p53R175H–HLA-A2 and 12-6–p53R175H–HLA-A2

complexes. The low resolution of the 1a2–p53R175H–HLA-A2 complex (3.00 Å) precluded identification of ordered waters with confidence.

To assess ligand-induced conformational changes in the TCRs, we determined the structures of 12-6 and 1a2 in unbound form to 2.36 and 1.83 Å resolution, respectively (TCR 38-10 did not crystallize) (Supplementary Table 3). Superpositions of the free and bound TCR 12-6 and 1a2 structures are shown in Supplementary Fig. 3. Superposition of the VαVβ domains of free 12-6 onto those of 12-6 in complex with p53R175H–HLA-A2

**Table 3 TCR CDR atomic contacts with peptide and MHC (percentage of contacts).**

| | | α chain | | | | β chain | | | |
|---|---|---|---|---|---|---|---|---|---|
| | | CDR1 | CDR2 | HV4 | CDR3 | CDR1 | CDR2 | HV4 | CDR3 |
| 12-6 | Peptide | 0 | 0 | 0 | 5 | 0 | 0 | 0 | 95 |
| | MHC | 5 | 42 | 0 | 30 | 5 | 3 | 0 | 13 |
| 38-10 | Peptide | 15 | 0 | 0 | 66 | 13 | 1 | 0 | 5 |
| | MHC | 21 | 16 | 0 | 37 | 13 | 5 | 0 | 8 |
| 1a2 | Peptide | 18 | 0 | 0 | 21 | 0 | 2 | 0 | 59 |
| | MHC | 22 | 30 | 0 | 47 | 0 | 0 | 0 | 2 |

Contacts were calculated between non-hydrogen atoms with a 4.0 Å distance cutoff.

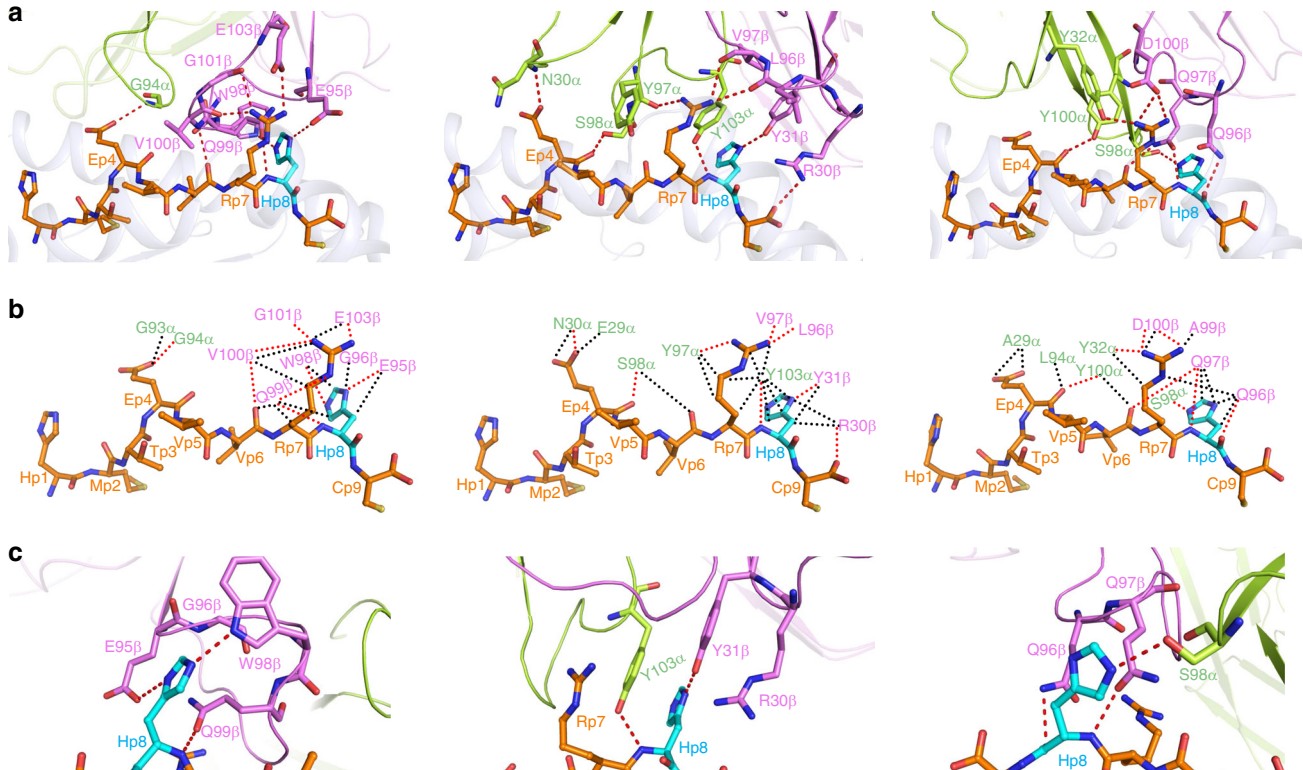

**Fig. 6 Interactions of TCRs with the p53R175H peptide. a** Interactions between 12-6 (left), 38-10 (center), and 1a2 (right) and the p53R175H peptide. The side chains of contacting residues are shown in stick representation with carbon atoms in green (TCR α chain), violet (TCR β chain), orange (p53R175H), or cyan (mutated P8 His), nitrogen atoms in dark blue, oxygen atoms in red, and sulfur atoms in yellow. Peptide residues are identified by one-letter amino acid designation followed by position (p) number. Hydrogen bonds are indicated by red dashed lines. **b** Comparison of interactions between 12-6 (left), 38-10 (center), and 1a2 (right) and the p53R175H peptide. Hydrogen bonds are red dotted lines and van der Waals contacts are black dotted lines. For clarity, not all van der Waals contacts are shown. **c** Close-up of interactions between 12-6 (left), 38-10 (center), and 1a2 (right) and P8 His.

revealed structural differences in the CDR3 loops (Fig. 7d). Whereas conformational adjustments in CDR3β were restricted mainly to shifts in side-chain orientation, CDR3α underwent a large movement (r.m.s.d. in α-carbon positions of 3.7 Å for residues 93–97), which allowed CDR3α Gly94 to hydrogen bond with P4 Glu and Arg65H, CDR3α Tyr95 to hydrogen bond with CDR3β Leu97 and contact the HLA-A2 α1 helix, and CDR3α Gln96 to hydrogen bond with Arg65H. CDR3α Tyr95 showed the largest individual displacement (6.5 Å in its α-carbon position). Ligand-induced conformational changes were also observed in the CDR3 loops of 1a2 (Fig. 7e). CDR3β underwent a rearrangement (r.m.s.d. in α-carbon positions of 1.8 Å for residues 96–102) that resulted in formation of four hydrogen bonds and 33 van der Waals contacts with P7 Arg and P8 His at the critical C-terminus of p53R175H. CDR3α experienced a

considerably larger movement (r.m.s.d. of 5.5 Å in α-carbon positions for residues 93–102), with CDR3α Glu96 undergoing an α-carbon displacement of 11.5 Å. This rearrangement allowed CDR3α to form a β-hairpin whose tip engages the HLA-A2 α1 helix and p53R175H, thereby optimizing TCR interactions with both MHC and peptide (Fig. 7f).

## Discussion

The development of T cell-based treatments that target tumor-specific neoantigens has become a central focus for cancer immunotherapy[1–9]. Neoantigen-directed therapies can be divided into two broad categories: neoantigen vaccines that seek to increase the number of neoantigen-specific T cells in vivo, and neoantigen-directed cell therapies in which neoantigen-specific

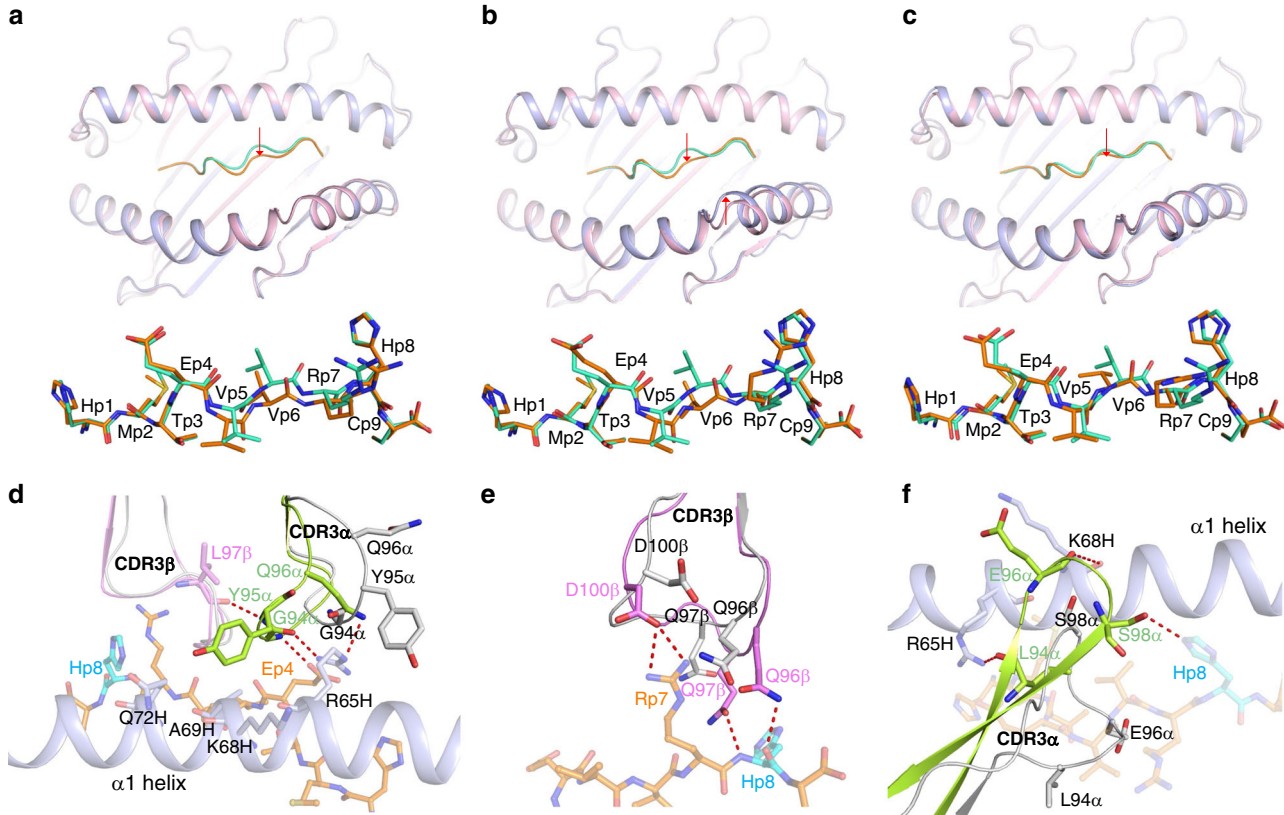

**Fig. 7 Conformational changes in p53R175H–HLA-A2 and TCRs after complex formation. a** (upper) Superposition of p53R175H–HLA-A2 in unbound form (peptide, green; HLA-A2, pink) and in complex with TCR 12-6 (peptide, orange; HLA-A2, light blue). Red arrow indicates region of structural shifts. (lower) Close-up showing residue shifts induced by TCR 12-6 binding. Note that where orange sticks appear behind green sticks, the peptide has shifted deeper into the pocket. **b** Similar superpositions showing peptide and HLA shifts on binding TCR 38-10. **c** Similar superpositions showing conformational changes induced by binding TCR 1a2. **d** Structural rearrangements in CDR3α of 12-6 induced by binding to p53R175H–HLA-A2 (bound 12-6, green or violet; unbound 12-6, gray; p53R175H, orange; HLA-A2, light blue). **e** Structural rearrangements in CDR3β of 1a2 induced by binding to p53R175H–HLA-A2. **f** Structural rearrangements in CDR3α of 1a2 induced by binding to p53R175H–HLA-A2.

T cells are provided to the patient to achieve this objective[7]. However, just how structurally different a neoantigen must be from its wild-type counterpart in order to overcome self-tolerance and elicit a T-cell response is not well understood. Bioinformatic profiling of cancer neoepitopes indicates that mutations at anchor positions which improve peptide binding to MHC molecules are associated with immunogenicity and tumor rejection[30,31]. However, most mutations in neoepitopes do not involve anchor residues and do not appreciably affect peptide binding to MHC. The p53R175H neoepitope studied here represents such a case.

Detailed comparison of the mutant p53R175H–HLA-A2 and wild-type p53–HLA-A2 structures revealed that their conformations differ only at the P8 mutation site, where one positively charged residue (histidine) replaces another (arginine). This substitution was sufficient to render the p53R175H peptide immunogenic in cancer patients[13,14]. Although replacement of histidine by arginine is conservative with respect to charge, these two amino acids differ markedly with respect to size and shape, which enables TCRs to distinguish between them. Assuming that p53 is expressed in the thymus and that the R175H mutation occurred after thymic development (i.e., during malignant transformation), the escape of p53R175H-specific T cells from negative selection is most likely explained by low affinity of the TCRs for wild-type p53 peptide. In support of this hypothesis, we were unable to detect any interaction of 12-6, 38-10, and 1a2 with p53–HLA-A2 by SPR, whereas these TCRs bound p53R175H–HLA-A2 with micromolar $K_{DS}$, which manifested as an overwhelming preference for histidine over arginine at P8 in

T-cell activation assays[13,14]. These results are consistent with previous structural studies of a unique melanoma neoepitope arising from a threonine-to-isoleucine mutation in a peptide derived from triose phosphate isomerase that produced only subtle changes in the binding surface for TCR[15,16]. Therefore, cancer neoantigens need differ only slightly from their wild-type counterparts for them to be immunogenic in patients. TCRs 12-6, 38-10, and 1a2 achieve high specificity for p53R175H by concentrating on the driver mutation at the C-terminal portion of the neoepitope, while avoiding extensive interactions with the N-terminal and central portions, which are shared with wild-type p53.

Structural studies of TCR–pMHC complexes involving common V segments and MHC alleles have revealed conservation of specific TCR–MHC interactions[32–34]. These conserved interactions, which occur between germline-encoded CDR1 and CDR2 loops and MHC, support the hypothesis that the canonical diagonal docking orientation of TCR on MHC, which is maintained in the TCR–p53R175H–HLA-A2 complexes, is the result of coevolution of TCR and MHC molecules. Surprisingly, however, while matches to at least one germline gene were found for TCRs 12-6, 38-10, and 1a2 among MHC class I-restricted TCRs in the structural database, inspection of the corresponding TCR–pMHC complexes indicated no matches to these TCRs in germline loop engagement of MHC. This unexpected lack of conserved TCR–MHC interactions applied even to TCRs restricted to HLA-A2, as well as to closely related MHC alleles such as HLA-B7 and HLA-B35. Therefore, considerably greater

flexibility exists for germline-encoded contacts between TCR and MHC than generally appreciated[32–34], supporting the notion that peptide "editing" can lead to variability in germline contacts with MHC[16,35]. This flexibility is exposed in the TCR–p53R175H–HLA-A2 complexes through the unusually high tilt of two of these TCRs over MHC and the shift of all three TCRs toward the peptide C-terminus relative to other MHC class I-restricted TCRs, a set which already exhibits a peptide C-terminal shift compared to MHC class II-restricted TCRs[36]. Also of note, no MHC class I-restricted TCRs were identified in the structural database that used the TRAV38-1 germline gene of TCR 38-10, and none that used the same TRAV/TRBV gene combinations as 12-6, 38-10, or 1a2. These newly described structures highlight that, in spite of the sizable number of TCR–pMHC complex structures determined to date, many functional germline interactions and docking geometries likely have yet to be revealed.

Considerable efforts have been made to engineer TCRs with improved affinity for cancer-associated antigens for use in ACT[37–39]. However, large gains in TCR affinity may lead to increased cross-reactivity[40], resulting in adverse clinical events[41]. In a striking case, an affinity-matured TCR targeting the MAGE-A3 melanoma antigen unexpectedly cross-reacted with an epitope from the muscle protein titin, causing cardiovascular toxicity and deaths[42]. Such off-target TCR recognition has prompted new structure-guided efforts to engineer therapeutic TCRs for enhanced specificity while maintaining optimal on-target affinity[41]. In one promising approach, the MART-1-specific TCR DMF5 was modified to promote stronger binding to the peptide portion of its pMHC ligand, which resulted in reduced cross-reactivity with MART-1 homologs[43]. An attractive feature of p53R175H-specific TCRs such as 12-6, 38-10, and 1a2 is that the parental T cells survived negative selection, thereby minimizing the possibility of cross-reactivity with self-antigens. However, we do not know whether these TCRs possess optimal on-target affinities. Accordingly, future efforts will be directed at rational design of p53R175H-specific TCRs to optimize on-target affinity without compromising neoepitope specificity. Alternatively, it was recently demonstrated that antigen-specific TCR function could be enhanced by structure-based mutations in the V$\alpha$ and V$\beta$ domains outside the CDR loops that increase the level of cell surface expression[44], an approach that could be attempted with the p53R175H-specific TCRs described here.

## Methods

**Protein preparation.** The isolation and characterization of p53R175H-specific TCRs 12-6, 38-10, and 1a2 from patients with epithelial cancers were described previously[13,14]. Soluble TCRs for affinity measurements and structure determinations were produced by in vitro folding from inclusion bodies expressed in *Escherichia coli*. Codon-optimized genes encoding the $\alpha$ and $\beta$ chains of these TCRs (TCR 12-6 residues 1–204 and 1–245; TCR 38-10 residues 1–211 and 1–244; TCR 1a2 residues 1–208 and 1–243, respectively) were synthesized (Supplementary Table 9) and cloned into the expression vector pET22b (GenScript). An interchain disulfide (C$\alpha$Cys158–C$\beta$Cys172 in TCR 12-6; C$\alpha$Cys165–C$\beta$Cys172 in TCR 38-10; C$\alpha$Cys162–C$\beta$Cys171 in TCR 1a2) was engineered to increase the folding yield of TCR $\alpha\beta$ heterodimers. The mutated $\alpha$ and $\beta$ chains were expressed separately as inclusion bodies in BL21(DE3) *E. coli* cells (Agilent Technologies). Bacteria were grown at 37 °C in LB medium to $OD_{600} = 0.6$–0.8 and induced with 1 mM iso-propyl-$\beta$-D-thiogalactoside. After incubation for 3 h, the bacteria were harvested by centrifugation and resuspended in 50 mM Tris-HCl (pH 8.0) containing 0.1 M NaCl and 2 mM EDTA. Cells were disrupted by sonication. Inclusion bodies were washed with 50 mM Tris-HCl (pH 8.0) and 5% (v/v) Triton X-100, then dissolved in 8 M urea, 50 mM Tris-HCl (pH 8.0), 10 mM EDTA, and 10 mM DTT. For in vitro folding, the TCR $\alpha$ (45 mg) and $\beta$ (35 mg) chains were mixed and diluted into 1 liter folding buffer containing 5 M urea, 0.4 M L-arginine-HCl, 100 mM Tris-HCl (pH 8.0), 3.7 mM cystamine, and 6.6 mM cysteamine. After dialysis against 10 mM Tris-HCl (pH 8.0) for 72 h at 4 °C, the folding mixture was concentrated 20-fold and dialyzed against 50 mM MES buffer (pH 6.0) to precipitate most incorrectly folded protein. Disulfide-linked TCR 12-6, 38-10, and 1a2 heterodimers were purified using sequential Superdex 200 (20 mM Tris-HCl (pH 8.0),

20 mM NaCl) and Mono Q (20 mM Tris-HCl (pH 8.0), 0–1.0 M NaCl gradient) FPLC columns (GE Healthcare).

Soluble HLA-A2 loaded with wild-type p53 peptide (HMTEVVRRC) or mutant p53R175H peptide (HMTEVV**R**HC) was prepared by in vitro folding. The HLA-A*0201 heavy chain (residues 1–275) and $\beta_2$-microglobulin (residues 1–99) were produced separately as inclusion bodies in BL21(DE3) *E. coli* cells transformed by pET26b containing the corresponding genes (53). Inclusion bodies were dissolved in 8 M urea, 50 mM Tris-HCl (pH 8.0), 10 mM EDTA, and 10 mM DTT. For in vitro folding, the HLA-A*0201 heavy chain (30 mg), $\beta_2$-microglobulin (30 mg), and wild-type or mutant p53 peptide (20 mg) (GenScript) were mixed and added dropwise to 1 liter of ice-cold folding buffer containing 5 M urea, 0.4 M L-arginine HCl, 100 mM Tris-HCl (pH 8.0), 5 mM EDTA, 3.7 mM cystamine, and 6.6 mM cysteamine. The folding mixture was dialyzed against distilled water for 24 h and then swapped into 10 mM Tris-HCl (pH 8.0) for 48 h at 4 °C. After concentration, correctly folded p53–HLA-A2 and p53R175H–HLA-A2 were purified using consecutive Superdex 200 (20 mM Tris-HCl (pH 8.0), 20 mM NaCl) and Mono Q columns (20 mM Tris-HCl (pH 8.0), 0–1.0 M NaCl gradient).

To prepare biotinylated HLA-A2, a 17-amino acid tag (GGGLNDIFEAQKIEW HE) was added to the C-terminus of the HLA-A*0201 heavy chain. The tagged p53–HLA-A2 and p53R175H–HLA-A2 proteins were produced as described above. Biotinylation was carried out using BirA biotin ligase (Avidity). Biotinylated protein was separated from excess biotin with a Superdex 200 column (50 mM sodium phosphate (pH 7.0), 100 mM NaCl).

**Crystallization and data collection.** For crystallization of TCR–p53R175H–HLA-A2 complexes, TCRs 12-6, 38-10, and 1a2 were each mixed with p53R175H–HLA-A2 in a 1:1 molar ratio and concentrated to 10 mg/ml. Crystals were obtained at room temperature by vapor diffusion in hanging or sitting drops. The 12-6–p53R175H–HLA-A2 complex crystallized in 20% (w/v) polyacrylic acid 5100, 0.1 M HEPES (pH 7.0), and 0.02 M MgCl$_2$. Crystals of the 38-10–p53R175H–HLA-A2 complex grew in 20% (w/v) polyethylene glycol (PEG) 3350 and 0.2 M ammonium tartrate dibasic. The 1a2–p53R175H–HLA-A2 complex crystallized in 12% (w/v) PEG 8000 and 0.1 M magnesium acetate. Crystals of unbound TCR 12-6 were obtained in 18% (w/v) PEG 2000, 0.2 M ammonium sulfate, and 0.1 M MES (pH 6.0). Unbound TCR 1a2 crystallized in 20% PEG 4000, 0.1 M MES (pH 6.0), and 0.2 M lithium sulfate. Crystals of p53–HLA-A2 and p53R175H–HLA-A2 grew in 15% PEG 8000, 0.1 M Tris-HCl (pH 8.0), and 0.2 M magnesium chloride. Before data collection, all crystals were cryoprotected with 20% (w/v) glycerol and flash-cooled. X-ray diffraction data for 12-6–p53R175H–HLA-A2, TCR 12-6, and 1a2–p53R175H–HLA-A2 were collected at beamline 23-ID-D of the Advanced Photon Source, Argonne National Laboratory. Data for p53–HLA-A2, p53R175H–HLA-A2, 38-10–p53R175H–HLA-A2, and TCR 1a2 were collected at beamline 19-BM. Diffraction data were indexed, integrated, and scaled using the program HKL3000[45]. Data collection statistics are shown in Supplementary Tables 1–3.

**Structure determination and refinement.** Before structure determination and refinement, all data reductions were performed using the CCP4 software suite[46]. Structures were determined by molecular replacement with the program Phaser[47] and refined with Phenix[48] and Refmac[49]. The models were further refined by manual model building with Coot[50] based on $2F_o$–$F_c$ and $F_o$–$F_c$ maps. The $\alpha$ chain of anti-EBV TCR RL42 (PDB accession code 3SJV)[51], the $\beta$ chain of anti-EBV TCR SB27 (2AK4)[52], and NLV–HLA-A2 (5D2L)[53] were used as search models with the CDRs and peptide removed to determine the orientation and position of the 12-6–p53R175H–HLA-A2 complex. The orientation and position parameters of unbound TCR 12-6, p53–HLA-A2, and p53R175H–HLA-A2 were obtained using the corresponding components of the 12-6–p53R175H–HLA-A2 complex.

Similarly, the $\alpha$ chain of an anti-HCV TCR (5YXN), the $\beta$ chain MART-1-specific TCR DMF4 (3QEQ)[54] and p53R175H–HLA-A2 with the CDRs and peptide removed were used as search models to determine the orientation and position of the 38-10–p53R175H–HLA-A2 complex. The $\alpha$ chain of preproinsulin-specific TCR 1E6 (3UTP)[55] and the $\beta$ chain of Nef-specific TCR T36-5 (3VXT)[56] with the CDRs removed were used as search models for molecular replacement to determine the structure of TCR 1a2. The structure of the 1a2–p53R175H–HLA-A2 complex was solved using TCR 1a2 and p53R175H–HLA-A2 as search models. Refinement statistics are summarized in Supplementary Tables 1–3. Contact residues were identified with the CONTACT program[46] and were defined as residues containing an atom 4.0 Å or less from a residue of the binding partner. The PyMOL program (https://pymol.org/) was used to prepare figures.

**Surface plasmon resonance analysis.** The interaction of TCRs 12-6, 38-10, and 1a2 with p53–HLA-A2 and p53R175H–HLA-A2 was assessed by surface plasmon resonance (SPR) using a BIAcore T100 biosensor at 25 °C. Biotinylated p53–HLA-A2 or p53R175H–HLA-A2 was immobilized on a streptavidin-coated BIAcore SA chip (GE Healthcare) at 3000 resonance units (RU). The remaining streptavidin sites were blocked with 20 μM biotin solution. An additional flow cell was injected with free biotin alone to serve as a blank control. For analysis of TCR binding, solutions containing different concentrations of 12-6, 38-10, or 1a2 were flowed sequentially over chips immobilized with p53–HLA-A2, p53R175H–HLA-A2, or

the blank. Both equilibrium and kinetic data were fitted with a 1:1 binding model using BIA evaluation 3.1 software.

**Computational structural analysis.** Previously determined structures of TCR complexes and their binding parameters were obtained from TCR3d (https://tcr3d.ibbr.umd.edu)[29]. The set of 151 MHC class I complex structures from TCR3d was filtered to retain only complexes with $\alpha\beta$TCRs, and to remove redundant complexes with identical TCR CDR loop and epitope sequences; this resulted in a set of 133 complex structures that was used for comparisons of docking orientations, positions, and contacts. Calculation of docking and incident angles was performed as previously described[22]. Calculation of $\Delta\Delta G$ for peptide point mutations was performed using Rosetta (release 2019.45), following a previously reported computational mutagenesis protocol[27]. This protocol was executed as a Rosetta Script, for which the code is available on Github on the Kortemme Lab ddg repository (https://github.com/Kortemme-Lab/ddg/), as part of the "alanine-scanning" protocol capture. The updated "REF15" scoring function in Rosetta[57] was used for packing and minimization during computational mutagenesis, and interaction $\Delta\Delta G$ were calculated with Rosetta's "interface" weights. Calculations of solvent-accessible surface areas were performed using the naccess program (http://wolf.bms.umist.ac.uk/naccess/). $S_c$ shape complementarity values were computed by the "sc" program in the CCP4 suite[46].

**Calculation of TCR centers.** TCR–pMHC complexes were oriented into a common reference frame centered at average C$\alpha$ atom position of MHC helices, with helix residues as defined previously[21], and rotated such that the $x$–$y$ plane is parallel with the helices, and the $x$-axis is parallel to peptide groove, with greater $x$ value corresponding to peptide C-terminus. Complex structures in this reference frame are downloadable from the TCR3d database (https://tcr3d.ibbr.umd.edu/downloads). TCR variable domain centers were calculated by taking centers of individual variable domains by average positions of S$\gamma$ atoms of conserved Cys residues (or C$\alpha$ atoms at corresponding positions where Cys residues are not present in the TCR), and then calculating the mean position of TCR V$\alpha$ and V$\beta$ centers. X position ($x$ pos) and y position ($y$ pos) values represent projections into the $x$–$y$ plane, and thus the MHC plane, of these centers.

**Reporting summary.** Further information on research design is available in the Nature Research Reporting Summary linked to this article.

## Data availability

Atomic coordinates and structure factors have been deposited in the Protein Data Bank under accession codes 6VR1 (p53–HLA-A2), 6VR5 (p53R175H–HLA-A2), 6VRM (TCR 12-6–p53R175H–HLA-A2), 6VRN (TCR 38-10–p53R175H–HLA-A2), 6VQO (TCR 1a2–p53R175H–HLA-A2), 6VTH (TCR 12-6), and 6VTC (TCR 1a2). Source data are provided with this paper. Other data are available from the corresponding author upon reasonable request.

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

## Acknowledgements

This work was supported by National Institutes of Health Grants GM126299 (to B.G.P.) and AI129893 (to R.A.M.). We thank Alexander Kolesnikov and Arjun Mishra for assistance with vector design and X-ray data collection, respectively. We also thank Paul F. Robbins and Steven A. Rosenberg (National Cancer Institute) for valuable discussions, as well as Drew C. Deniger (National Cancer Institute) for providing the corrected amino acid sequence of the TCR 1a2 $\alpha$ chain. Results in this report are based on work performed at both Structural Biology Center and GM/CA beamlines at the Advanced Photon Source of Argonne National Laboratory, operated by UChicago Argonne, LLC, for the U.S. Department of Energy, Office of Biological and Environmental Research under contract DE-AC02-06CH11357. Identification of commercial materials and equipment does not imply recommendation nor endorsement by the National Institute of Standards and Technology, nor does it imply that the material or equipment identified is the best available for the purpose.

## Author contributions

D.W., D.T.G. and R.G. performed the experiments and data analyses. B.G.P. and R.A.M. conceived and supervised the project. All authors prepared the paper.

## Competing interests

The authors declare no competing interests.
