## [Peer Review File · Nature Communications]

Reviewer #1 (Remarks to the Author):

Wu et al. characterized comprehensive structural studies for oligoclonal TCRs specific to a shared epitope of the p53 cancer neoantigen restricted to HLA-A2 molecules. They determined almost a whole set of crystal structures of free HLA-A2s, free TCRs and their complexes. They found important points for different and common features of docking modes and detail TCR/HLA/peptide interactions, which is highly informative for rational design of high-affinity and high-specificity TCRs. This paper is interesting and has potential to advance our knowledge for a field of immunotherapy. I have some minor comments.

Comments

1. They described in detail every interaction in Result section, however, it may be difficult for readers to follow. Each subparagraph may had better point out the main findings at the beginning.
2. Discussion, Line 365. "This unexpected lack of conserved.... HLA-B35" Are there any correlation whether the patients have such B alleles?
3. The authors should show the kinetic parameters of SPR analysis and may discuss these with structural dynamics upon the binding.
4. The authors should show the structural comparison of whole domain of TCRs at the free and complex forms.
5. Supplementary table 5 is important and thus should be shown as one of the main Tables.

Reviewer #2 (Remarks to the Author):

This manuscript offers the delineation of structural information relating to the presentation of a driver mutation in p53 oncogene which makes cancer cells visible to the adaptive immune system. The manuscript is concise and clearly written. There is no need to alter the presentation significantly. Published literature around this field is adequately referenced and used critically to put the results into the correct context.

The conclusions made by the authors are convincingly justified by the evidence presented. There is no hyperbole or unnecessary digression into speculation. The methodology is sufficiently explained that an appropriately trained individual can reproduce the work reported.

The results presented are novel and should be of interest to researchers who design cancer treatments with Adoptive T-Cell therapies.

There are a few comments that need to be examined by the authors, as suggestions for improvement to their manuscript:

- line 68: I think 'affect' should be changed to 'effect' as in 'cause'
- line 135: An extra word in 'extending the entire' to read 'extending along the entire' would make the sentence clearer
- line 139: The preference for P2 and P9 ligands were quoted from previously published research. Looking up the reference provided showed they were both second best! Perhaps an adjusted comment would address this.
- line 169: The 133 PDB entries are listed in the Supplementary material. Please refer to the correct table in the text.
- line 272-274: I would turn this statement around and say the skewing is a result of the positioning. I know it's semantics, but the fact remains that the TCR is chasing after the binding epitope and has to position itself in the right place and accept whatever skewing occurs. I don't insist on this point of view, though.
- line 282: Please state which atom in Tyr98beta is contacting peptide H8, like the other contacts

mentioned in this line.

- para starting at line 284: Was the modelling of H8R subjected to any refinement other than graphical adjustment? I am thinking specifically of either idealising the geometry without reference to the X-ray terms, or maybe some thermodynamics 'relaxation' to allow an 'ideal' fit, from the point of view of the energetics of the binding. Would the Rosetta exercise do this automatically? I personally use geometric idealisation followed by PISA estimation of the interface energy, and each residues contribution.

A second point here is the fact that A 3K-cal/Mol is about the random energy available from the environment at physiological temperature. The estimated stabilisation energies are lower than that, but the SPR binding measurement is clearly favourable. I would think the coordinated binding of all the peptide residues would add to a significant sum. But I also fret over the surface complementarity, which would be quite important at the interface. Would the authors comment on this point, please. Quoting the SC statistic would also be useful.

- line 302: P6 Val is pushed 'deeper' into the binding groove upon TCR docking. This is part of the characteristic bulge in the peptide and is well clear of the floor of the groove. This volume occasionally hosts water molecules or buffer or cryo-protectant molecules. When the TCR docks, some of these 'interlopers' would be pushed out. Was anything like this observed in this case?

- line 341: Charge conservation is not enough, as you show quite elegantly. Size and shape are just as important, and these are significantly different, between His and Arg, to cause mayhem. Similar 'conservative' mutations have been observed elsewhere which cause as much disruption, or even worse.

Supplementary Information: Statistics Tables S1, S2 and S3

- Please add 1 row at the top of each table indicating the PDB entry of each data set.
- Please add 1 row showing CC1/2 statistics, if available.
- Supp. Table 2 figures in the first column (12-6 Complex with pMHC) do not match with validation report 6VRM. Please check.

PDB Validation Reports:

- 6VQO: Chain descriptions are not correct. Please adjust. The validation report indicates significant twinning. The R-factors are relatively high compared with other entries in the database, perhaps because no account of twinning had been taken during refinement. If twin refinement has indeed been used, please indicate the twin laws and the fraction of each, data to be entered in the statistics table. An update of coordinates may also be necessary.

- 6VR1: No chain descriptions for Molecules 1 or 2

- 6VRM: Chain B is described as 'tumor suppressor from p53' when it is beta2 micro-globulin. Chain E 'is a protein', when it should be TCR beta-chain

- 6VRN: Chain descriptions need attention as above

- 6VTC: Molecule 1 is called 'T-cell receptor 1a2', Molecule 2 has no description. A better description would be 'T-cell receptor 1a2 alpha chain' and 'T-cell receptor 1a2 beta chain'

- 6VTH: Molecule 1 is called 'T-cell receptor 12-6', Molecule 2 has no description. A better description would be 'T-cell receptor 12-6 alpha chain' and 'T-cell receptor 12-6 beta chain'

Reviewer #3 (Remarks to the Author):

Wu and colleagues present data and structures describing, in detail, the molecular the interactions of the MHC molecule, HLA-A2, with the shared neoantigen p53 R175H. While HLA-A2 presenting the WT p53 peptide is not immunogenic and has not been associated with TCR binding that leads to activation, the mutated p53 peptide (bound by HLA-A2) has been found in multiple patients to be recognized by T cells, providing evidence of a "public" neoantigen that can be the target for more widespread immunotherapies. The authors provide the structural and biophysical basis for the recognition of the HLA-A2, p53 R175H antigen. The authors investigate 3 previously identified

TCRs from various patients (12-6, 1a2, and 38-10). These three TCRs all have orientations that are deviant from the majority of HLA-A2:TCR structures from the PDB. The authors then go on to detail the molecular underpinnings of these structures and complexes, including contacts via peptide and MHC, where it is suggested that C-terminally biased TCR binding is preferred. These findings are of high importance to our basic understanding of neoantigen recognition, but will also provide the basis for further engineering for use in immunotherapy and thus is highly appropriate for publication in Nature Communications.

We suggest some editorial changes for the manuscript, mostly consolidation of the molecular descriptions (which are highly detailed in the text) and a minor error in one of the figures:

1. The information found in figures 4-7 can be better consolidated. While the points for each of the figures is understood, I believe some of these can be combined. More importantly, the text regarding these figures can be substantially trimmed. While all of the details are interesting, all presented together could impact the ability of the reader to easily glean the really important points from the structures.

2. In figure 1, although the SPR trace look good, there is a discrepancy between (f) top, kinetic data, and bottom, equilibrium fit. It appears that the titration from the above is lacking the concentration that would have been tested with the green color. However, this green concentration is the final point in the bottom panel. Please explain or fix this in the graph.

3. Tables 2 and 3 are overwhelming. While it is a very important piece of data found within the structure, I do not feel this belongs in the main text. This would be more appropriate in the supplement.

4. Please increase the size of the labels in the figures and consider changing the color of the light green to slightly darker in Figs. 5, 6 and 7.

Erin Adams

Title: Structural basis for oligoclonal T cell recognition of a shared p53 cancer neoantigen
Daichao Wu, D. Travis Gallagher, Ragul Gowthaman, Brian G. Pierce and Roy A. Mariuzza

Reply to Reviewer #1

We are grateful to the reviewer for his/her appreciation of our study and for calling our attention to the following points (changes in the manuscript text file are indicated with yellow highlighting):

1. They described in detail every interaction in Result section, however, it may be difficult for readers to follow. Each subparagraph may better point out the main findings at the beginning.

Response: As suggested, we revised the section headings to give the main messages as follows:

-TCRs are highly specific for mutant p53 peptide.

-Differences between p53–HLA-A2 and p53R175H–HLA-A2 are confined to mutation site.

-TCRs are shifted towards C-terminus of p53R175 peptide.

-V α dominates contacts with MHC.

-TCRs target p53 driver mutation.

-Conformational changes optimize TCR–pMHC interactions.

2. Discussion, Line 365. “This unexpected lack of conserved.... HLA-B35” Are there any correlation whether the patients have such B alleles?

Response: Patient 36, from which TCRs 12-6 and 38-10 were isolated, expressed HLA-A*0201 and HLA-A*2402 (ref. 14). No mention was made of HLA-B alleles. Patient 1, from which TCR 1a2 was isolated, expressed HLA-A*0201 (ref. 15). No information was given for HLA-B alleles.

3. The authors should show the kinetic parameters of SPR analysis and may discuss these with structural dynamics upon the binding.

Response: We carried out kinetic analysis of the binding of TCR 12-6 to p53R175H–HLA-A2 (Fig. 1d). Kinetic parameters were $k_{\text{on}} = 2.5 \times 10^4 \text{ M}^{-1}\text{s}^{-1}$ and $k_{\text{off}} = 0.032 \text{ s}^{-1}$, corresponding to a K_D of 1.3 μM , which is in good agreement with the K_D from equilibrium analysis (1.1 μM). For TCRs 38-10 and 1a2, k_{on} and k_{off} were too rapid to be reliably measured (Fig. 1e, f), as is often the case for TCR–pMHC interactions. We have added this information to the revision (lines 124-127): *Kinetic parameters (on- and off-rates) for the binding of TCR 12-6 to p53R175H–HLA-A2 were $k_{\text{on}} = 2.5 \times 10^4 \text{ M}^{-1}\text{s}^{-1}$ and $k_{\text{off}} = 0.032 \text{ s}^{-1}$, corresponding to a K_D of 1.3 μM (Fig. 1d), which is in close agreement with the K_D from equilibrium analysis (1.1 μM). For TCRs 38-10 and 1a2, k_{on} and k_{off} were too rapid to be measured reliably (Fig. 1e, f).*

4. The authors should show the structural comparison of whole domain of TCRs at the free and complex forms.

Response: We have added a figure showing superpositions of free and bound TCRs 12-6 and 1a2 (lines 318-319):

*Superpositions of the free and bound TCR 12-6 and 1a2 structures are shown in **Supplementary Fig. 3.***

5. Supplementary table 5 is important and thus should be shown as one of the main Tables.

Response: As requested, we have moved Supplementary Table 5 to the main tables (now Table 2).

Title: Structural basis for oligoclonal T cell recognition of a shared p53 cancer neoantigen
Authors: Daichao Wu, D. Travis Gallagher, Ragul Gowthaman, Brian G. Pierce and Roy A. Mariuzza

Reply to Reviewer #2

We thank the reviewer for his/her careful reading of our manuscript and for raising important points to address (changes in the manuscript text file are indicated with yellow highlighting):

- line 68: I think 'affect' should be changed to 'effect' as in 'cause'

Response: We have made this correction (line 68).

- line 135: An extra word in 'extending the entire' to read 'extending along the entire' would make the sentence clearer

Response: We have added “along” to this phrase (line 138).

- line 139: The preference for P2 and P9 ligands were quoted from previously published research. Looking up the reference provided showed they were both second best! Perhaps an adjusted comment would address this.

Response: Thank you for pointing this out. We have corrected this sentence according to the amino acid frequencies listed in Table 1 of ref. 20 (lines 142-144):

*Methionine and cysteine are among the most common residues at primary anchor positions P2 (Leu > Thr > Met ~ Val > Ile) and P9 (Val > Ile > Thr > Ala > Cys > Leu), and are known to confer high affinity for HLA-A*02:01 (20).*

- line 169: The 133 PDB entries are listed in the Supplementary material. Please refer to the correct table in the text.

Response: We have inserted a reference to Supplementary Table 4 (line 174).

- line 272-274: I would turn this statement around and say the skewing is a result of the positioning. I know it's semantics, but the fact remains that the TCR is chasing after the binding epitope and has to position itself in the right place and accept whatever skewing occurs. I don't insist on this point of view, though.

Response: We have rewritten this sentence as suggested (lines 266-268):

However, the TCRs share a pronounced skewing toward the peptide C-terminus (Supplementary Table 4) as a result of positioning one or both of their CDR3 loops directly over P8 His (Fig. 6a, b, c).

- line 282: Please state which atom in Tyr98beta is contacting peptide H8, like the other contacts mentioned in this line.

Response: We have added this information (lines 274-277):

Further selectivity for the mutant p53 peptide arises from hydrogen bonds with the P8 His side chain: 12-6 Glu95 β O ϵ 2–N δ 1 P8 His, 12-6 Trp98 β N ϵ 1–N ϵ 2 P8 His, 38-10 Tyr31 β OH–N δ 1 P8 His, and 1a2 Ser98 α O γ –N ϵ 2 P8 His (Supplementary Table 6).

- para starting at line 284: Was the modelling of H8R subjected to any refinement other than graphical adjustment? I am thinking specifically of either idealising the geometry without reference to the X-ray terms, or maybe some thermodynamics 'relaxation' to allow an 'ideal' fit, from the point of view of the energetics of the binding. Would the Rosetta exercise do this automatically? I personally use geometric idealisation followed by PISA estimation of the interface energy, and each residues contribution.

A second point here is the fact that A 3K-cal/Mol is about the random energy available from the environment at physiological temperature. The estimated stabilisation energies are lower than that, but the SPR binding measurement is clearly favourable. I would think the coordinated binding of all the peptide residues would add to a significant sum. But I also fret over the surface complementarity, which would be quite important at the interface. Would the authors comment on this point, please. Quoting the S_c statistic would also be useful.

Response: We thank the reviewer for the questions regarding the modeling and the impact of the H8R substitution. We have added more information regarding the modeling in Rosetta in the Methods section. Our modeling in Rosetta included packing of the mutant side chain in Rosetta to optimize its conformation prior to energetic scoring, and did not include additional minimization. This protocol was used previously by our team to model impact of substitutions in TCR–pMHC interfaces (now referenced in the Results). To remove ambiguity regarding the predicted impact and the modeling of this substitution, we have removed Supplementary Figure 3, which included superposed pMHC structures rather than modeled substitutions. We have expanded the corresponding paragraph in the Results to include the discussion of the S_c statistic, as well as energetic terms from Rosetta, to explore possible energetic and structural mechanisms underlying the dramatic observed specificity, noted below (lines 278-298):

*To assess possible structural defects leading to TCR affinity loss for the p53 revertant peptide, we calculated TCR–pMHC shape complementarity statistics (S_c) for the X-ray and modeled p53 revertant interfaces. S_c values for p53175H interfaces are 0.72, 0.64, and 0.69 for 12-6, 38-10, and 1a2 respectively, commensurate with other MHC class I TCR–pMHC structures in the TCR3d database (29), while they are 0.69, 0.57, and 0.71 for p53 revertants and the same respective TCRs, indicating loss of shape complementarity for the 12-6 and 38-10 TCR interfaces, and less predicted effect on the shape complementarity of the 1a2 interface. To further investigate the mechanistic basis of peptide specificity for these TCRs, the individual Rosetta scoring function terms comprising the predicted $\Delta\Delta G$ values noted above and in **Supplementary Table 7** were obtained (**Supplementary Table 8**). This revealed that loss of favorable van der Waals interactions dominated the change in predicted binding affinity for the 12-6 TCR, whereas disruptions of side chain–side chain hydrogen bond interactions involving P8 His were primarily responsible for predicted 38-10 and 1a2 TCR affinity losses.*

- line 302: P6 Val is pushed 'deeper' into the binding groove upon TCR docking. This is part of the characteristic bulge in the peptide and is well clear of the floor of the groove. This volume occasionally hosts water molecules or buffer or cryo-protectant molecules. When the TCR docks, some of these 'interlopers' would be pushed out. Was anything like this observed in this case?

Response: We thank the reviewer for asking about water under the peptide. There are two adjacent water sites under peptide residues P6–P8 that are occupied in both the wild-type p53–HLA-A2 and unbound mutant p53R175H–HLA-A2 structures. Both sites remain occupied in the 38-10–p53R175H–HLA-A2 and 12-6–p53R175H–HLA-A2 complexes. The low resolution of the 1a2–

p53R175H–HLA-A2 complex (3.00 Å) prevented us from assigning waters with confidence. We have added three sentences to Results to note these findings (lines 311-315):

There are two adjacent water molecules under peptide residues P6–P8 in the wild-type p53–HLA-A2 and unbound mutant p53R175H–HLA-A2 structures. These waters are retained in the 38-10–p53R175H–HLA-A2 and 12-6–p53R175H–HLA-A2 complexes. The low resolution of the 1a2–p53R175H–HLA-A2 complex (3.00 Å) precluded identification of ordered waters with confidence.

• line 341: Charge conservation is not enough, as you show quite elegantly. Size and shape are just as important, and these are significantly different, between His and Arg, to cause mayhem. Similar 'conservative' mutations have been observed elsewhere which cause as much disruption, or even worse.

Response: We agree completely, and have modified the text to make this important point (lines 348-353):

Detailed comparison of the mutant p53R175H–HLA-A2 and wild-type p53–HLA-A2 structures revealed that their conformations differ only at the P8 mutation site, where one positively charged residue (histidine) replaces another (arginine). This substitution was sufficient to render the p53R175H peptide immunogenic in cancer patients (13, 14). Although replacement of histidine by arginine is conservative with respect to charge, these two amino acids differ markedly with respect to size and shape, which enables TCRs to distinguish between them.

Supplementary Information: Statistics Tables S1, S2 and S3

• Please add 1 row at the top of each table indicating the PDB entry of each data set.

Response: We have added PDB accession codes.

• Please add 1 row showing CC_{1/2} statistics, if available.

Response: We have added a row to Supplementary Tables 1, 2, and 3 to give the CC_{1/2} values for each dataset, both overall and for the highest resolution shell.

• Supp. Table 2 figures in the first column (12-6 Complex with pMHC) do not match with validation report 6VRM. Please check.

Response: The reviewer's observation led us to re-refine and re-deposit the 12-6–p53R175H–HLA-A2 complex structure (6VRM). The *B*-values in the relatively mobile outward domains of this structure are somewhat high, giving an overall average *B* of 78. However, the mean *B* for the peptide is 63, and the binding region is entirely clear and well-ordered. There are no significant changes in atom positions. Statistics in the Supplementary Tables are now in agreement with the PDB Validation Reports for all structures.

PDB Validation Reports:

• 6VQO: Chain descriptions are not correct. Please adjust. The validation report indicates significant twinning. The *R*-factors are relatively high compared with other entries in the database, perhaps because no account of twinning had been taken during refinement. If twin refinement has indeed been used, please indicate the twin laws and the fraction of each, data to be entered in the statistics table. An update of coordinates may also be necessary.

Response: We thank the reviewer for observing the evidence for twinning in the 6VQO structure. We have re-refined the structure with account for twinning, and the *R*-values are significantly lower. The new structure is very similar, differing by an overall r.m.s.d. of 0.124 Å. There are no significant changes in the peptide-binding regions. We have revised Supplementary Table 2 accordingly, now reporting the twin laws and twin fraction.

- 6VR1: No chain descriptions for Molecules 1 or 2

Response: We have added chain descriptions.

- 6VRM: Chain B is described as 'tumor suppressor from p53' when it is beta2 micro-globulin. Chain E 'is a protein', when it should be TCR beta-chain

Response: We have added chain descriptions.

- 6VRN: Chain descriptions need attention as above

Response: We have added chain descriptions.

- 6VTC: Molecule 1 is called 'T-cell receptor 1a2', Molecule 2 has no description. A better description would be 'T-cell receptor 1a2 alpha chain' and 'T-cell receptor 1a2 beta chain'

Response: We have added chain descriptions.

- 6VTH: Molecule 1 is called 'T-cell receptor 12-6', Molecule 2 has no description. A better description would be 'T-cell receptor 12-6 alpha chain' and 'T-cell receptor 12-6 beta chain'

Response: We have added chain descriptions.

Title: Structural basis for oligoclonal T cell recognition of a shared p53 cancer neoantigen
Authors: Daichao Wu, D. Travis Gallagher, Ragul Gowthaman, Brian G. Pierce and Roy A. Mariuzza

Reply to Reviewer #3

We thank Professor Adams for her appreciation of our study, and for her suggestions to improve the manuscript (changes in the manuscript text file are indicated with yellow highlighting).

1. The information found in figures 4-7 can be better consolidated. While the points for each of the figures is understood, I believe some of these can be combined. More importantly, the text regarding these figures can be substantially trimmed. While all of the details are interesting, all presented together could impact the ability of the reader to easily glean the really important points from the structures.

Response: Because we are reporting multiple structures, Figs. 4-7 each have at least six panels, making consolidation difficult. However, we agree that some details about structures can be removed from the text to improve readability. We have trimmed the text accordingly without compromising the main points. In addition, we revised the section headings to better convey the key messages:

-TCRs are highly specific for mutant p53 peptide.

-Differences between p53–HLA-A2 and p53R175H–HLA-A2 are confined to mutation site.

-TCRs are shifted towards C-terminus of p53R175 peptide.

-V α dominates contacts with MHC.

-TCRs target p53 driver mutation.

-Conformational changes optimize TCR–pMHC interactions.

2. In figure 1, although the SPR trace look good, there is a discrepancy between (f) top, kinetic data, and bottom, equilibrium fit. It appears that the titration from the above is lacking the concentration that would have been tested with the green color. However, this green concentration is the final point in the bottom panel. Please explain or fix this in the graph.

Response: Thank you for noticing this discrepancy. We have corrected the color of the highest concentration titration in Fig. 1f (upper) to green.

3. Tables 2 and 3 are overwhelming. While it is a very important piece of data found within the structure, I do not feel this belongs in the main text. This would be more appropriate in the supplement.

Response: As suggested, we have moved Tables 2 and 3 to Supplementary Information.

4. Please increase the size of the labels in the figures and consider changing the color of the light green to slightly darker in Figs. 5, 6 and 7.

Response: As requested, we have increased the size of the labels in Figs. 5, 6 and 7. However, we prefer to use light green for the TCR α chains because this is the same green that we used for the TCR α chains in Figs. 3 and 4. In addition, we used dark green for the unbound p53R175H peptide in Fig. 7a, b, c.

REVIEWERS' COMMENTS:

Reviewer #2 (Remarks to the Author):

I thank the authors for having positively responded to all the points I raised. The adjustments to the manuscript are satisfactory, and I consider the manuscript ready for publication.